

# How to account for behavioral states in step-selection analysis: a model comparison

Jennifer Pohle[1], Johannes Signer[2], Jana A. Eccard[3], Melanie Dammhahn[4] and Ulrike E. Schlägel[1]

[1] Institute of Biochemistry and Biology, University of Potsdam, Potsdam, Germany
[2] Wildlife Sciences, Faculty of Forest Sciences and Forest Ecology, University of Goettingen, Göttingen, Germany
[3] Animal Ecology, Institute of Biochemistry and Biology, University of Potsdam, Potsdam, Germany
[4] Department of Behavioural Biology, University of Münster, Münster, Germany

## ABSTRACT

Step-selection models are widely used to study animals' fine-scale habitat selection based on movement data. Resource preferences and movement patterns, however, often depend on the animal's unobserved behavioral states, such as resting or foraging. As this is ignored in standard (integrated) step-selection analyses (SSA, iSSA), different approaches have emerged to account for such states in the analysis. The performance of these approaches and the consequences of ignoring the states in step-selection analysis, however, have rarely been quantified. We evaluate the recent idea of combining iSSAs with hidden Markov models (HMMs), which allows for a joint estimation of the unobserved behavioral states and the associated state-dependent habitat selection. Besides theoretical considerations, we use an extensive simulation study and a case study on fine-scale interactions of simultaneously tracked bank voles (*Myodes glareolus*) to compare this HMM-iSSA empirically to both the standard and a widely used classification-based iSSA (*i.e.*, a two-step approach based on a separate prior state classification). Moreover, to facilitate its use, we implemented the basic HMM-iSSA approach in the R package HMMiSSA available on GitHub.

## INTRODUCTION

Combining animal movement and environmental data, step-selection analysis (SSA) and its extension, the integrated step-selection analysis (iSSA), build a popular framework for studying animals' fine-scale habitat selection, while also taking the movement capacity of the animal into account (*Fortin et al., 2005*; *Forester, Im & Rathouz, 2009*; *Avgar et al., 2016*; *Northrup et al., 2022*). Essentially, they explain the animals' space use based on possible preferences for or avoidance of environmental features, considering spatial limitations that the animals' movement process imposes on the features' availability. ISSAs have successfully been applied, for example, to analyze elk response to roads (*Prokopenko, Boyce*

Corresponding author
Jennifer Pohle,
jennifer.pohle@uni-potsdam.de

& *Avgar, 2017*), to study the effects of artificial nightlight on predator–prey dynamics of cougars and deer (*Ditmer et al., 2021*), and to model space use of Cape vultures in the context of wind energy development (*Cervantes et al., 2023*). Besides conventional habitat use, SSA has also proven suitable for detecting interactions such as avoidance or attraction between simultaneously tracked individuals (*Schlägel et al., 2019*).

SSA and iSSA typically use a conditional logistic regression for case-control designs to compare the characteristics of observed, *i.e.*, *used* steps against the covariates of alternative steps *available* at a given time point. Here, a step is the straight-line segment connecting two consecutive locations sampled at regular time intervals and is usually described by the step length and turning angle, *i.e.*, the directional change (*Fortin et al., 2005*). The covariates usually correspond to features of the steps' end point, *e.g.*, vegetation or snow cover (*Stratmann et al., 2021*), but can also refer to characteristics along the step, *e.g.*, the presence of roads on the path (*Prokopenko, Boyce & Avgar, 2017*). What is considered to be available at a given time point depends on the assumptions made about the animals' movement capacities and/or typical movement patterns. This usually translates into assumptions about the animals' step length and turning angle distributions (*e.g.*, gamma and von Mises distributions). Tentative parameter estimates for these distributions can be estimated from the observed steps. These estimates are usually used in SSA and iSSA to randomly draw available (also called control) steps. They are, however, biased because movement and thus the observed steps are influenced by habitat selection (*Forester, Im & Rathouz, 2009*). While SSA ignores this potential bias, a correction for the tentative parameters can be estimated within the iSSA (*Avgar et al., 2016*; *Fieberg et al., 2021*). Other estimation approaches might circumvent this problem arising from the use of conditional logistic regression (*e.g.*, *Schlägel, Merrill & Lewis, 2017*), leading to the correct estimates of the underlying model immediately, but require a more manual implementation. In this article we generally use the term iSSA for approaches where the movement parameters are explicitly estimated together with the habitat selection coefficients.

While (i)SSAs seem suitable in numerous instances, it has recently been argued that fine-scale habitat selection and resource requirements might depend on the animal's behavioral modes such as resting or foraging (illustrated in Fig. 1). Ignoring such states in the analysis might thus lead to biased results and misleading conclusions (*Roever et al., 2014*; *Suraci et al., 2019*). With telemetry-based location data, however, the underlying behavioral states are usually unobserved. Therefore, it has been suggested to first classify the movement data into different states, *e.g.*, based on hidden Markov models (HMMs, *Zucchini, MacDonald & Langrock, 2016*), and to split the step observations accordingly into state-specific data sets, which can then be used to fit state-specific (i)SSAs in a second step (*Roever et al., 2014*; *Karelus et al., 2019*; *Picardi et al., 2022*). This two-step approach, hereafter named TS-(i)SSA, accounts for the unobserved state structure and is convenient as it can be based on existing software implementations (*e.g.*, R-package moveHMM for HMM-based state separation and R-package amt for subsequent step selection analysis; *Michelot, Langrock & Patterson, 2016*; *Signer, Fieberg & Avgar, 2019*). It has, however, two major drawbacks. First, the state classification is purely based on movement patterns without considering habitat selection. Thus, habitat selection and selection-independent movement processes can be

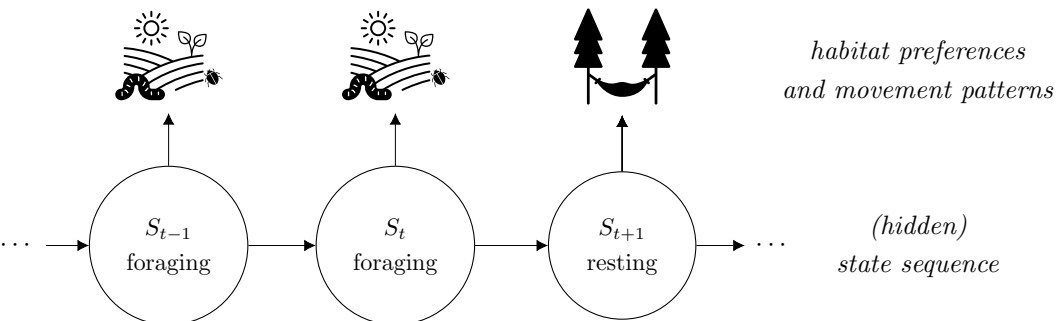

**Figure 1  Illustration of how behavioral states can affect animals' habitat selection and movement patterns.** The state "foraging" is related to search for food such as small insects in an open landscape, while the state "resting" is associated with a retreat in its shelter. Usually, the behavioral states are unobserved, thus hidden, and serially correlated. This structure corresponds to the basic dependence structure of a hidden Markov step-selection model.

confounded when defining the states. This can affect the validity of the state classification and can lead to a bias in the estimated movement and selection parameters (*Prima et al., 2022*). Second, the uncertainty in the HMM state classification is completely ignored in the follow-up (i)SSA. Possible misclassification can again lead to biased movement and (habitat) selection coefficients. Furthermore, confidence intervals and standard *p*-values are no longer reliable as the uncertainty of both the HMM parameter estimation and the state classification are not taken into account. Consequently, also the TS-iSSA might lead to biased results and misleading conclusions. How serious this is in practice, however, has rarely been quantified. *Prima et al. (2022)* evaluated a population-level version of the TS-iSSA in a simulation study and found good classification and prediction performances in the scenarios considered, but also biased parameter estimates. By focusing on the population level, however, they did not provide results on the variation, uncertainty quantification and estimation accuracy of the individually fitted TS-iSSA models.

The above mentioned problems can be avoided by combining step selection models and HMMs in a single model to allow for a joint estimation of the underlying state, habitat selection and movement processes. First proposed by *Nicosia et al. (2017)* and recently extended by *Klappstein, Thomas & Michelot (2023)*, this hidden Markov model step selection function (HMM-SSF) approach renders a prior state classification unnecessary. All model parameters can be jointly estimated using a case-control Markov-switching conditional logistic regression framework. Since in this article we only consider the approach involving explicit estimation of the movement parameters, we refer to the method as HMM-iSSA. Similar to *Klappstein, Thomas & Michelot (2023)* we use a numerical maximum likelihood estimation, but constrain all parameters to their natural parameter space to avoid problems in model interpretation (*e.g.*, a negative shape parameter for an assumed gamma distribution for step length). For state decoding, *i.e.*, to assign a state to each step observation *after* parameter estimation, we consider the well-known

Viterbi algorithm. It computes the most likely state sequence underlying the data given the estimated model parameters.

The aim of this article is two-fold. First, we provide a broad overview of the HMM-iSSA framework by discussing the underlying movement model, its relation to alternative approaches (iSSA, TS-iSSA, HMMs) and, most importantly, its practical implementation, which we further facilitate through release as an R package. Second, we investigate whether and to what extent either the complete neglect of underlying states in the analysis (iSSA) or their incorporation through a prior HMM-based state classification (TS-iSSA) affects the estimation results compared to the HMM-iSSA approach. For this, we use an extensive simulation study to compare the estimation and, if applicable, classification performance of iSSAs, TS-iSSAs and HMM-iSSAs in three state-switching scenarios. Thereby, we showcase different ways in which behavioral states could influence the animals' movement decisions. A supplementary simulation covers a scenario without underlying state-switching. We further compare the three approaches in a case study on fine-scale interactions of bank voles (*Myodes glareolus*), which are small ground-dwelling rodents. Using a movement data set of synchronously tracked individuals, as analyzed in *Schlägel et al. (2019)*, we test whether HMM-iSSAs can detect meaningful biological states and whether they provide new insights into interactions, such as attraction, avoidance, or neutrality towards other conspecifics compared to iSSAs. Portions of this text were previously published as part of a preprint (https://doi.org/10.48550/arXiv.2304.12964).

## METHODS

### Hidden Markov step-selection model

We use $\{\mathbf{x}_{0,1}, \mathbf{x}_{0,2}, \ldots, \mathbf{x}_{0,T}\}$ to denote the sequence of two-dimensional animal locations observed at regular time intervals, which forms the observed movement track. We here use the subscript "0" to distinguish these *used* locations from *control* locations (introduced below). Conditional on the previous location $\mathbf{x}_{0,t-1}$, a step from the current location $\mathbf{x}_{0,t}$ to the next location $\mathbf{x}_{0,t+1}$, is characterized by its step length $l_{0,t+1}$, *i.e.*, the straight-line distance between the two consecutive locations, and its turning angle $\alpha_{0,t+1}$, *i.e.*, the directional change. The corresponding covariate vector $\mathbf{Z}_{0,t+1}$ stores the feature values of the step, and we use $\mathbf{Z}$ to denote the collection of covariate values for all possible locations in the given spatial domain $D$ in which the animal moves.

In the hidden Markov step-selection model (*Nicosia et al., 2017*; *Klappstein, Thomas & Michelot, 2023*; *Prima et al., 2022*), we assume the observed steps to be driven by an underlying hidden state sequence $\{S_1, S_2, \ldots, S_T\}$ with $N$ discrete states. Thus, each state variable $S_t$ at time $t$ can take one of $N$ state values ($S_t \in \{1, \ldots, N\}$). These states serve as proxies for the unobserved behavioral modes of the animal that influence its movement and habitat selection (illustrated in Fig. 1). We assume the state sequence to be a homogeneous $N$-state Markov chain, characterized by its transition probabilities $\gamma_{ij} = \Pr(S_t = j | S_{t-1} = i)$ to switch from state $i$ to state $j$, summarized in the $N \times N$ transition probability matrix $\Gamma$, and the initial state distribution $\delta$ which contains the probabilities to start in a certain state.

Each state $i$ ($i = 1, \ldots, N$) is associated to a state-dependent density $f_i$ generating the next location. Its functional form is similar to the basic step-selection model (*Forester,*

*Im & Rathouz, 2009*), but with movement and habitat selection parameters being state-dependent. Thus, conditional on locations $\mathbf{x}_{0,t-1}$ and $\mathbf{x}_{0,t}$, and covariates $\mathbf{Z}$, the current state $S_t = i$ determines the following distribution for a step to location $\mathbf{x}_{0,t+1}$:

$$f_i(\mathbf{x}_{0,t+1}|\mathbf{x}_{0,t},\mathbf{x}_{0,t-1},\mathbf{Z};\boldsymbol{\theta}_i,\boldsymbol{\beta}_i) = \frac{\overbrace{\phi(\mathbf{x}_{0,t+1}|\mathbf{x}_{0,t},\mathbf{x}_{0,t-1};\boldsymbol{\theta}_i)}^{\text{movement kernel}} \cdot \overbrace{\omega(\mathbf{Z}_{0,t+1};\boldsymbol{\beta}_i)}^{\text{selection function}}}{\underbrace{\int_{\tilde{\mathbf{x}}\in D} \phi(\tilde{\mathbf{x}}|\mathbf{x}_{0,t},\mathbf{x}_{0,t-1};\boldsymbol{\theta}_i) \cdot \omega(\tilde{\mathbf{Z}};\boldsymbol{\beta}_i)d\tilde{\mathbf{x}}}_{\text{normalizing constant}}}, \qquad (1)$$

where $\tilde{\mathbf{Z}}$ in the integral denotes the covariate vector of location $\tilde{\mathbf{x}}$. The density consists of three components: (i) The movement kernel $\phi(\cdot)$ describes the space use in a homogeneous landscape and is usually defined in terms of step length $l_{0,t+1}$ (*e.g.*, gamma distribution) and turning angle $\alpha_{0,t+1}$ (*e.g.*, von Mises distribution). The corresponding state-dependent parameters for state $i$ are summarized in the movement parameter vector $\boldsymbol{\theta}_i$; (ii) The movement kernel is weighted by the selection function $\omega(\cdot)$ which indicates a possible selection for or against the covariates in $\mathbf{Z}_{0,t+1}$. It is usually assumed to be a log-linear function of the (state-dependent) selection coefficient vector $\boldsymbol{\beta}_i$,

$$\omega(\mathbf{Z}_{0,t+1};\boldsymbol{\beta}_i) = \exp\left(\mathbf{Z}_{0,t+1}^{\top}\boldsymbol{\beta}_i\right),$$

where a positive selection coefficient indicates preference for, and a negative coefficient avoidance of a corresponding covariate; (iii) The integral in the denominator ensures that $f_i$ integrates to one. Usually, it is analytically intractable and must therefore be approximated, for example, using numerical integration methods. We provide an example of state-dependent step-selection densities in a 2-state scenario in Fig. S1.

There are important relations between the hidden Markov step-selection model and two alternative movement models: (i) If all states share the same parameters, *i.e.*, $\boldsymbol{\theta}_1 = \ldots = \boldsymbol{\theta}_N$ and $\boldsymbol{\beta}_1 = \ldots = \boldsymbol{\beta}_N$, or if the number of states is set to one, *i.e.*, $N = 1$, the model reduces to the basic step-selection model without state-switching (*Forester, Im & Rathouz, 2009*); (ii) If all selection coefficients are equal to zero, *i.e.*, $\boldsymbol{\beta}_1 = \ldots = \boldsymbol{\beta}_N = \mathbf{0}$, the model reduces to a basic movement HMM(*Langrock et al., 2012*, *Patterson et al., 2017*) with state-dependent step length and turning angle distributions as implied by the movement kernel $\phi(\cdot)$ but without habitat selection. These relations are very convenient, for example in the context of model comparison and model selection, as it allows the use of standard tests or information criteria to select between these three candidate models.

We can simplify the step-selection density $f_i$ by assuming step lengths follow a distribution from the exponential family (with support on non-negative real numbers, *e.g.*, a gamma distribution) and turning angles follow either a uniform or von Mises distribution with fixed mean (*Avgar et al., 2016*; *Nicosia et al., 2017*). In this case, the product of the movement kernel $\phi(\cdot)$ and the exponential selection function $\omega(\cdot)$ is proportional to a single log-linear function of the corresponding model parameters and $f_i$ reduces to:

$$f_i(\mathbf{x}_{0,t+1}|\mathbf{x}_{0,t},\mathbf{x}_{0,t-1},\mathbf{Z};\boldsymbol{\theta}_i,\boldsymbol{\beta}_i) = \frac{\exp\left(\mathbf{C}_{0,t+1}^{\top}\boldsymbol{\theta}_i + \mathbf{Z}_{0,t+1}^{\top}\boldsymbol{\beta}_i - \log(l_{0,t+1})\right)}{\int_{\tilde{\mathbf{x}}\in D} \exp\left(\tilde{\mathbf{C}}^{\top}\boldsymbol{\theta}_i + \tilde{\mathbf{Z}}^{\top}\boldsymbol{\beta}_i - \log(\tilde{l})\right)d\tilde{\mathbf{x}}}, \qquad (2)$$

where $\tilde{\mathbf{C}}$, $\tilde{\mathbf{Z}}$ and $\tilde{l}$ depend on location $\tilde{\mathbf{x}}$. The vector $\mathbf{C}_{0,t+1}$ can be interpreted as a movement covariate vector that contains different step length and turning angle terms. Its exact form depends on the chosen step length and turning angle distributions (Table S1, see also *Avgar et al., 2016*, *Nicosia et al., 2017*). For example, for gamma-distributed step length and von-Mises-distributed turning angles with either mean zero or $\pi$, we have $\mathbf{C}_{0,t+1} = (\log(l_{0,t+1}), -l_{0,t+1}, \cos(\alpha_{0,t+1}))^\top$. The corresponding state-dependent movement coefficient vector is $\boldsymbol{\theta}_i = (k_i - 1, r_i, \kappa_i)^\top$ with $k_i$ and $r_i$ being the shape and rate parameter of the gamma-distribution belonging to state $i$, respectively, and $|\kappa_i|$ being the state-dependent concentration parameter of the von-Mises distribution($\kappa_i < 0$ indicates a von Mises distribution with mean $\pi$; *Nicosia et al., 2017*). Thus, in this reduced representation of the step-selection density $f_i$, the parameterisation of the movement kernel might differ from the commonly used parameterisation of the corresponding step and angle distributions (*e.g.*, $k_i - 1$ instead of $k_i$), but there is a direct relationship between the two (Tables S1 and S2). The negative log step length included in the exponential function is necessary to correctly represent the movement kernel in a Cartesian coordinate system.

The reduced form of $f_i$ is very convenient. Justified by the law of large numbers, it allows for a joint parameter estimation of the state, movement and selection parameters based on a Markov-switching conditional logistic regression for case-control designs with $M$ control, *i.e.*, available, locations per observed, *i.e.*, used, location (*Nicosia et al., 2017*, see also *Avgar et al. 2016* for step-selection models without state-switching). This forms the basis for the standard HMM-iSSA.

## Parameter estimation

The HMM-iSSA workflow is similar to the one of the standard iSSA. For each observed step, we choose $M$ control steps, usually using a suitable parametric proposal distribution for step length and turning angle, respectively, and extract the corresponding habitat and movement covariate values. This builds the case-control data set. The model parameters are then estimated using a Markov-switching conditional logistic regression, *i.e.*, a conditional logistic regression in which the regression coefficients depend on an underlying latent Markov chain. As *Klappstein, Thomas & Michelot (2023)*, we use the forward algorithm, which is well-known especially in the context of HMMs (*Zucchini, MacDonald & Langrock, 2016*) to efficiently evaluate the corresponding likelihood. This allows for a numerical maximum likelihood estimation based on standard optimization procedures such as *nlm* in R (*R Core Team, 2022*). Afterwards, it is possible to decode the states, for example, using the Viterbi algorithm (*Viterbi, 1967*), which calculates the most likely sequence of states given the fitted model and the case-control data.

More precisely, for each step from location $\mathbf{x}_{0,t}$ to $\mathbf{x}_{0,t+1}$ ($t = 2, \ldots, T-1$), we create a choice set $\tilde{\mathbf{x}}_{t+1} = \{\mathbf{x}_{0,t+1}, \mathbf{x}_{1,t+1}, \ldots, \mathbf{x}_{M,t+1}\}$ that includes the observed and the $M$ control locations for the end point of the step. Usually, the control steps are randomly drawn from a suitable proposal distribution for step length and turning angle (*Forester, Im & Rathouz, 2009*). However, it is also possible to use a grid or a mesh (*Arce Guillen et al., 2023*). Here the devil is in the detail, as depending on the sampling procedure, the interpretation of the models' movement coefficients might differ (see Section S2). The interpretation of the

selection coefficients, however, remain unaffected if $M$ is chosen large enough to provide stable results.

In this case-control setting, we model the state-dependent choice probability $p_{0,t,i}$ of choosing the step to the observed location $\mathbf{x}_{0,t+1}$ from the choice set $\tilde{\mathbf{x}}_{t+1}$ given the current state $S_t = i$, as:

$$p_{0,t,i}(\mathbf{x}_{0,t+1}|\tilde{\mathbf{x}}_{t+1}, \mathbf{C}, \mathbf{Z}; \boldsymbol{\theta}_i, \boldsymbol{\beta}_i) = \frac{\exp\left(\mathbf{C}_{0,t+1}^{\top}\boldsymbol{\theta}_i + \mathbf{Z}_{0,t+1}^{\top}\boldsymbol{\beta}_i\right)}{\sum_{m=0}^{M}\exp\left(\mathbf{C}_{m,t+1}^{\top}\boldsymbol{\theta}_i + \mathbf{Z}_{m,t+1}^{\top}\boldsymbol{\beta}_i\right)}, \tag{3}$$

with $\mathbf{C}_{m,t+1}$ and $\mathbf{Z}_{m,t+1}$ being the movement and habitat covariate vectors belonging to location $\mathbf{x}_{m,t+1}$ for $m = 0, \ldots, M$. This case-control step-selection probability is closely related to direct numerical integration, which offers an alternative way to approximate the step-selection density $f_i$. Note that in contrast to Eq. (2), the term $-\log(l_{.,t+1})$ does not appear in Eq. (3) because we change from Cartesian to polar coordinates when sampling control steps (*i.e.*, step lengths and turning angles).

We derive the likelihood of the Markov-switching conditional logistic regression by plugging $p_{0,t,i}$ into the HMM likelihood (*Zucchini, MacDonald & Langrock, 2016*),

$$\mathcal{L}(\boldsymbol{\theta}, \boldsymbol{\beta}; \tilde{\mathbf{x}}_3, \tilde{\mathbf{x}}_4, \ldots, \tilde{\mathbf{x}}_T, \mathbf{C}, \mathbf{Z}) = \boldsymbol{\delta}^{\top}\mathbf{P}(\tilde{\mathbf{x}}_3)\boldsymbol{\Gamma}\mathbf{P}(\tilde{\mathbf{x}}_4)\boldsymbol{\Gamma}\cdots\boldsymbol{\Gamma}\mathbf{P}(\tilde{\mathbf{x}}_T)\mathbf{1}, \tag{4}$$

where $\mathbf{P}(\tilde{\mathbf{x}}_{t+1}) = \mathrm{diag}(p_{0,t,1}, \ldots, p_{0,t,N})$ is a diagonal matrix including the state-dependent step-selection probabilities, $\boldsymbol{\Gamma}$ and $\boldsymbol{\delta}$ are the transition probability matrix and the initial distribution of the underlying Markov chain, respectively, and $\mathbf{1}$ is an $N$-dimensional vector of ones. We can then estimate the model parameters using a numerical maximization of the log-likelihood (for details, see *Zucchini, MacDonald & Langrock, 2016*). In our implementation, we restrict the movement parameters to always remain in their natural parameter space, *e.g.*, the shape and rate parameters of the gamma distribution are constraint to values greater than zero.

For initialization, the numerical maximization requires a set of starting values for the model parameters. To avoid ending up in a local maximum of the log-likelihood, it is necessary to test several sets of starting values, for example by randomly drawing initial values for each model parameter. We discuss this in more detail in Section S3.

## Two-step approach

The TS-iSSA is based on the same idea as the HMM-iSSA. However, the TS-iSSA relies on a *prior* classification of the movement data into different movement states. Thus, in a first step, an $N$-state HMM with state-dependent step length and turning angle distributions as defined for the movement kernel is fitted to the data, *e.g.*, using a gamma distribution for step length and a von Mises distribution for turning angles. Then, the Viterbi algorithm is used to assign each observed step to one of the $N$ HMM movement states. Alternatively, local state decoding can be used. In the second step, state-specific (i)SSAs are applied to the $N$ state-specific data sets using a case-control design and conditional logistic regression (*e.g.*, *Roever et al., 2014*; *Karelus et al., 2019*). The control steps for the state-specific case-control data sets are usually sampled based on the respective state-dependent HMM step length and turning angle distributions.

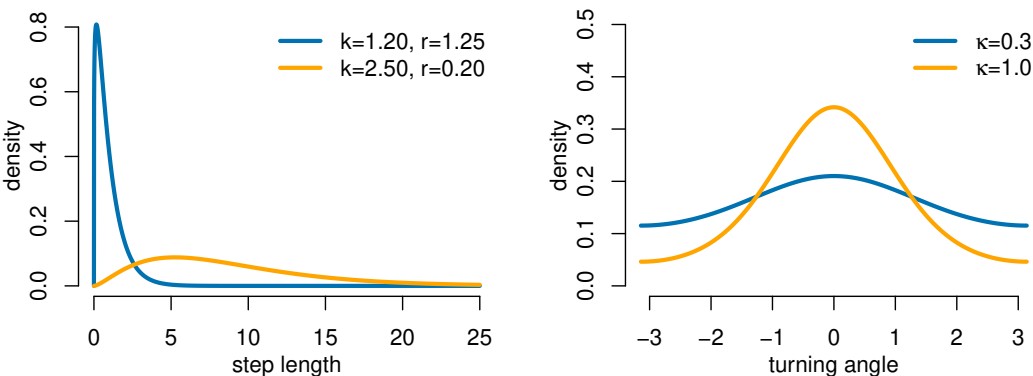

**Figure 2** **Gamma and von Mises distributions for step length and turning angle, respectively, as used in the simulation study (movement kernel).** Parameters are denoted by *k* (shape), *r* (rate) and *κ* (concentration). The distributions for state 1 are shown in blue and for state 2 in orange, except for Scenario 2 in which both states share the same movement kernel (in orange).

## Simulation study

We used a simulation study with three state-switching scenarios to evaluate the performance of our HMM-iSSA approach and to demonstrate possible consequences of either ignoring the underlying latent states in the traditional iSSA or ignoring the uncertainty of prior state-decoding in the TS-iSSA. For each scenario, we generated movement data from a hidden Markov step-selection model with 2 states and state transition probabilities $\gamma_{11} = \gamma_{22} = 0.9$. A realization of a Gaussian random field with covariance $\sigma^2 = 1$ and range $\phi = 10$, computed using the function *grf* from the R-package *geoR* (*Ribeiro Jr et al., 2022*), served as the habitat covariate **Z** (Fig. S2). For the movement kernel, we used state-dependent gamma and zero-mean von Mises distributions to model step length and turning angle, respectively (Fig. 2).

Table 1 summarizes the movement and selection parameters for each of the three simulation scenarios. Scenario 1 is chosen to represent a typical inactive-active scenario in which the first state ("inactive" state) is associated to small step length, less directive movement and no selection, while the second state ("active" state) corresponds to larger step length, more directed movement and attraction to the landscape feature **Z**. The second and the third scenarios cover the rather extreme cases in which either the selection or the movement parameters are shared across states: In Scenario 2 ("switching preferences"), the two states only differ in their selection patterns with avoidance of the feature in state 1, and attraction to the feature in state 2. In Scenario 3 ("HMM"), only the movement patterns differ across states while there is no selection for or against the landscape feature in either state. This corresponds to a basic movement HMM. To check the robustness of the HMM-iSSA, we additionally covered a fourth scenario without state-switching in which the data are generated based on a standard step-selection model (Section S5.2). Furthermore, to check the influence of the spatial variation in the habitat feature on the estimation results, we also considered a second landscape feature map with weaker spatial heterogeneity (Section S5.3).

**Table 1** **Overview of the hidden Markov step-selection model parameters for each simulation scenario.** The selection coefficients $\beta_i$ for state $i$ ( $i = 1, 2$) belong to the selection function of the model. The movement kernel parameters are the shape $k_i$ and rate $r_i$ of gamma distribution for step length and the concentration parameter $\kappa_i$ of the von-Mises distribution for turning angle. Scenario 4 does not include any state-switching and is covered in the Supplementary Material.

| | state 1 | | | | state 2 | | | |
| | select. fun. | Movement kernel | | | select. fun. | Movement kernel | | |
| Scenario | $\beta_1$ | $k_1$ | $r_1$ | $\kappa_1$ | $\beta_2$ | $k_2$ | $r_2$ | $\kappa_2$ |
|---|---|---|---|---|---|---|---|---|
| 1 (active-inactive) | 0.00 | 1.20 | 1.25 | 0.30 | 2.00 | 2.50 | 0.29 | 1.00 |
| 2 (switching preferences) | −2.00 | 2.50 | 0.29 | 1.00 | 2.00 | 2.50 | 0.29 | 1.00 |
| 3 (HMM) | 0.00 | 1.20 | 1.25 | 0.30 | 0.00 | 2.50 | 0.29 | 1.00 |
| 4 (iSSA, Supp. Mat.) | 2.00 | 2.50 | 0.29 | 1.00 | – | – | – | – |

In each of the 100 simulation runs per scenario, we simulated movement paths of length $T = 1000$ from the corresponding hidden Markov step-selection model and then applied 2-state HMM-iSSAs, 2-state TS-iSSAs and iSSAs to corresponding case-control data sets with randomly drawn control steps using a uniform distribution for turning angle and a proposal gamma distribution for step length, respectively (Section S2). To check whether the parameter estimates converge to stable values, we considered $M = 20$, $M = 100$ and $M = 500$ control locations per observed step. For model selection purposes, we further estimated the parameters of a 2-state HMM-iSSA with movement but without habitat covariates. This corresponds to a basic movement HMM, but fitted to the same case-control data as the candidate models. After parameter estimation, we computed AIC and BIC values (*Burnham & Anderson, 2002*) and standard *p*-values for the estimated selection coefficients. For TS-iSSAs and HMM-iSSAs we further computed the state missclassification rate, *i.e.*, the percentage of states that were not correctly classified using the Viterbi algorithm. These metrics and the parameter estimates were used to evaluate the estimation and classification accuracy of the candidate models, and to assess the performance of standard model selection procedures.

## Case study on bank vole interactions

To illustrate the use of HMM-iSSAs on empirical data, we applied them to existing movement data of synchronously tracked bank vole individuals (*Myodes glareolus*) as analyzed in *Schlägel et al. (2019)*. The original sample included 10 group-level replicates with 4 animals each, combining animal personalities (not part of this study) and sex. All bank vole females were sexually mature, but not gravid or lactating. All bank vole males were mature. Individuals within a replicate were synchronously tracked in fenced quadratic outdoor enclosures of 2,500 m² for 3–5 days using collars with small radio telemetry transmitters (1.1 g, BD-2C, Holohil Systems Ltd., Canada) and a system of automatic receiving units (Sparrow systems, USA). For bank vole individuals tracked under natural conditions, the estimated home range sizes were on average $2,029$m² with a core area of 549m² (*Schirmer et al., 2019*). Thus, the size of the enclosures allowed the individuals to express their natural movement and space use. The original sample

size allowed for potential technical failures of tracking equipment or escapes of single individuals from semi-natural enclosures, which indeed reduced the sample size to 8 populations with 2 males and 1–2 females each and a total of 28 animals. The tracking system produced 6-minute location data. Due to daily system maintenance, locations were missing for approximately one hour per day. Otherwise, movement paths were complete. Depending on the replicate, this resulted in 602–1,200 locations per individual split into 3–5 bursts of around 23 h each. After the experiment animals were captured from the enclosures, transmitter collars were removed, and animals were returned to the wild at their individual capture location. For more details on the bank vole capture and location data, see *Schlägel et al. (2019)*. Details on animal care approvals and research permissions are provided in the Section 6.1.

To study interactions between the bank vole individuals, *i.e.*, attraction, avoidance or neutral behavior towards each other, *Schlägel et al. (2019)* applied SSAs to each individual of each replicate, respectively, using occurrence estimates of the conspecifics as covariates. The occurrence estimate of an individual provides a map of the individual's space use during a certain time window, indicating areas of higher and lower probability of occurrence during that time period. It is estimated from the discrete sample of observed locations through kriging (*Fleming et al., 2016*). To account for the movement of individuals, occurrence estimates were computed using a rolling time window (here 4 h).

The analysis focused on interactions between males and females: Males were expected to mainly show attraction towards females, while females could show any of the three behaviors depending on their reproductive state (*Schlägel et al., 2019*). The authors suggested that the relatively large number of non-significant selection coefficients, especially found for male interactions with females, might be caused by unobserved mixtures of different underlying behavioral modes. Bank voles are polyphasic with resting phases of approximately 3h and active phases of approximately 1h following each other (*Mironov, 1990*). We therefore applied 2-state HMM-iSSAs to the same data to investigate (i) if the state-switching model is capable to detect meaningful biological states, and (ii) if we find different significant selection coefficients using the state-switching approach.

For each individual, we used a 2-state HMM-iSSA with state-dependent gamma distributions for step length, and uniform distribution for turning angle, respectively. Occurrence estimates of each conspecific within the same replicate were used as covariates for the selection part of the model (*Schlägel et al., 2019*). As interactions were analyzed within replicates, *i.e.*, only for animals tracked simultaneously in same enclosure, a nested approach controlled for potential confounds such as differences among enclosures. We did not include a resource covariate, as vegetation was sufficiently homogeneous within enclosures. Furthermore, we chose to use $M = 500$ control steps per used step, as preliminary analysis with increasing values for $M$ provided stable results for this choice (Fig. S9). Thus, with $M = 500$ control steps, the corresponding selection covariate vector for individual $k$ at time $t$ and locations $\mathbf{x}_{m,t}$, $m = 1, \ldots, 500$, was given by $\mathbf{Z}_{k,m,t} = (\{O_{-k,m,t}\})$, where $\{O_{-k,m,t}\}$ denotes the set of occurrence estimates of the respective conspecifics withing the same replicate. The corresponding movement covariate vector was $\mathbf{C}_{k,m,t} = (\log(l_{k,m,t}), -l_{k,m,t})$. Parameters were then estimated using 50 sets of

Table 2 Percentage of simulation runs in which the selection coefficients are estimated to be significantly different from zero at a significance level of $\alpha = 0.05$, for each scenario and fitted model, respectively.

| model | no. cont. | Scen. 1 | | Scen. 2 | | Scen. 3 | |
|---|---|---|---|---|---|---|---|
| | | $\beta_1 = 0$ | $\beta_2 = 2$ | $\beta_1 = -2$ | $\beta_2 = 2$ | $\beta_1 = 0$ | $\beta_2 = 0$ |
| iSSA | 20 | | 100 | | 57 | | 16 |
| | 100 | | 100 | | 57 | | 17 |
| | 500 | | 100 | | 58 | | 17 |
| TS-iSSA | 20 | 40 | 100 | | 58 | 5 | 6 |
| | 100 | 42 | 100 | | 57 | 4 | 6 |
| | 500 | 39 | 100 | | 57 | 4 | 5 |
| HMM-iSSA | 20 | 2 | 100 | 100 | 100 | 4 | 5 |
| | 100 | 3 | 100 | 100 | 100 | 2 | 5 |
| | 500 | 1 | 100 | 100 | 100 | 5 | 6 |

random starting values. For model comparison, we further applied corresponding iSSAs (no state-switching) and HMM-iSSAs without selection covariates (*i.e.*, HMM approximations; no selection) to the same case-control data set for each individual. For the 2-state TS-iSSAs (prior state-classification), control steps were sampled using the estimated state-dependent distributions of the movement HMMs fitted in the first step of the TS-iSSA.

Both the simulation and the case study were implemented in R (*R Core Team, 2022*). We used custom code for HMMs and HMM-iSSAs and the clogit-function from the R package survival (*Therneau, 2020*) for the iSSA and corresponding parts of the TS-iSSA. The R code is available in Supplementary Information 2.

## RESULTS

### Simulation study

Overall, the HMM-iSSA performed very well across all simulation scenarios and did not produce any evident bias even in the extremer Scenarios 2 ("state-switching preferences") and 3 ("HMM"; Fig. 3, Tables S3 and S4). The TS-iSSA was able to detect two suitable states in both scenarios with state-dependent movement kernels (Scenarios 1 and 3), although there was a small but evident bias for some parameters, for example, for the selection coefficients in scenario 1 (0.18 in state 1, −0.13 in state 2 for $M = 500$), and for the shape parameter in scenario 3 (0.12 in state 2 for $M = 500$). For Scenario 1 ("active-inactive"), this is also reflected in the rather large percentage of significant selection coefficients across the simulation runs in state 1 (39–42% at a significance level of $\alpha = 0.05$, Table 2), although the true coefficient is equal to zero. Thus, in contrast to the HMM-iSSA, the *p*-values of the TS-iSSA are not reliable in this active-inactive setting.

The iSSA is by its nature unable to distinguish between the underlying states and thus, did not recover the true underlying parameters in either scenario. Especially in Scenario 2 ("switching preferences"), the iSSA selection coefficients were estimated close to zero and the associated *p*-values would misleadingly indicate no selection for or against the landscape feature in 42% − 43% of the simulation runs (Table 2). Note that the TS-iSSA

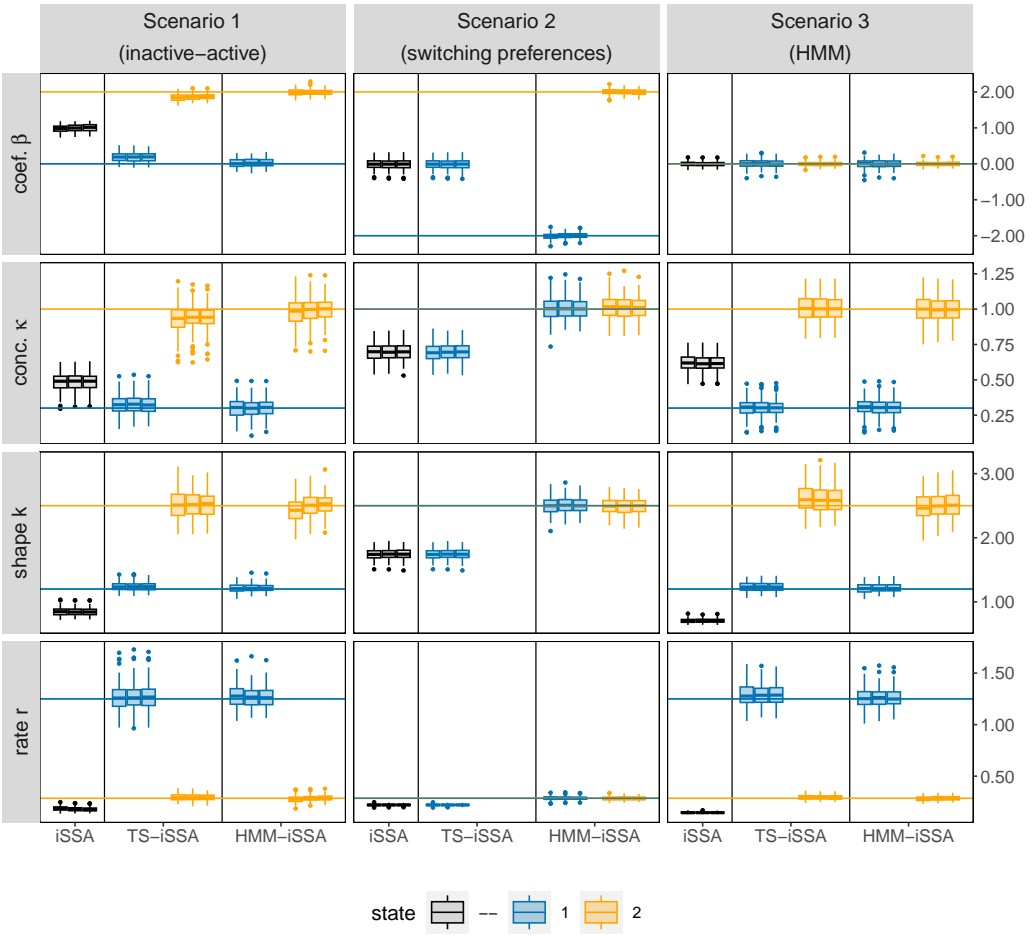

**Figure 3** **Boxplots of the parameter estimates across the 100 simulation runs for each applied method, simulation scenario and number of control locations $M$, respectively.** The rows refer to the estimated selection coefficient ($\beta$), the shape and rate of the gamma-distribution for step length and the concentration parameter ($\kappa$) of the von Mises distribution for turning angle, respectively. The columns refer to the three different simulation scenarios. For each method (iSSA, TS-iSSA and HMM-iSSA) and state (state 1: blue, state 2: orange, no state differentiation: black), the three adjacent boxplots refer the use of $M = 20$, $M = 100$ and $M = 500$ control locations per used location for the parameter estimation. Note that in Scenario 2, the TS-iSSA is naturally not capable to distinguish between two states as both share the same movement kernel. Thus, there are only results for a single state.

produced similar results to the iSSA in this scenario, since the inherent HMM classification was not able to distinguish between states that share the same movement kernel, and therefore all steps were classified to belong to the same state.

For all three simulation scenarios, the number of available steps $M$ only slightly affected the estimation results in this simulation exercise, especially the results for $M = 100$ and $M = 500$ are very similar (Fig. 3). Thus, the results seem to be stable. The HMM-iSSA with $M = 500$ available steps achieved the lowest missclassification rate (Table 3). As the data in Scenario 3 were simulated from an HMM, the HMM classification was equally accurate
**Table 3** **Mean missclassification rate (in %) with standard deviation in parentheses across the 100 simulation runs for each scenario and fitted state-switching model, respectively.** The missclassification rate is calculated as the percentage of states incorrectly classified using the Viterbi sequence. The lowest missclassification rate for each scenario is highlighted in bold face.

| | HMM-iSSA | | | |
| scenario | 20 | 100 | 500 | HMM |
|---|---|---|---|---|
| 1 (active-inactive) | 4.05 (0.83) | 3.76 (0.82) | **3.70** (0.79) | 5.93 (1.47) |
| 2 (switching preferences) | 2.12 (0.53) | 2.00 (0.52) | **1.94** (0.50) | 49.01 (4.38) |
| 3 (HMM) | 2.49 (0.49) | 2.42 (0.51) | **2.38** (0.55) | 2.39 (0.53) |

**Table 4** **Percentage of simulation runs in which the three candidate models are selected by either AIC or BIC for each simulation scenario and number of control points used for model fitting, respectively.** The cells belonging to the true underlying model are highlighted using bold face.

| | | AIC | | | BIC | | |
| Scenario | no. cont. | iSSA | HMM[*] | HMM-iSSA | iSSA | HMM[*] | HMM-iSSA |
|---|---|---|---|---|---|---|---|
| | 20 | 0 | 0 | **100** | 0 | 0 | **100** |
| Scen. 1 | 100 | 0 | 0 | **100** | 0 | 0 | **100** |
| | 500 | 0 | 0 | **100** | 0 | 0 | **100** |
| | 20 | 2 | 0 | **98** | 2 | 0 | **98** |
| Scen. 2 | 100 | 2 | 0 | **98** | 2 | 0 | **98** |
| | 500 | 2 | 0 | **98** | 2 | 0 | **98** |
| | 20 | 0 | **88** | 12 | 0 | **100** | 0 |
| Scen. 3 | 100 | 0 | **91** | 9 | 0 | **100** | 0 |
| | 500 | 0 | **86** | 14 | 0 | **100** | 0 |

**Notes.**
  *Here, HMM corresponds to the HMM-iSSA fitted without selection covariates, which provides an approximation to the movement HMM.

in this scenario. Overall, the HMM-iSSA clearly outperformed the other candidate models in its estimation and classification performance in all scenarios.

As the TS-iSSA involves an a-priori HMM classification, it does not provide a proper maximum likelihood value. It is therefore not possible to calculate corresponding AIC or BIC values for model selection. Thus, we only considered iSSAs without state-switching, HMMs without selection (fitted to the same case-control data sets) and HMM-iSSAs as candidate models to evaluate information-criteria based model selection in this modeling framework. For Scenario 1 and 2, AIC and BIC performed very well and selected the true underlying model in 100 and 98% of the simulation runs, respectively (Table 4). In Scenario 3 ("HMM"), the AIC tended to select the true HMM model in most of the cases but occasionally selected the more complex HMM-iSSA ($9 - 14$% of the runs), while the BIC again selected the correct model in all simulation runs.

Overall, the simulation runs with lower spatial variation in the landscape variable produced similar results (Section S5.3). However, the lower spatial variation reduces the influence of the habitat selection function on space use. Therefore, the variance in the estimates slightly increased, the HMM missclassification rate decreased in Scenario 1 and the HMM-iSSA missclassification rate increased in Scenario 2. In the supplementary

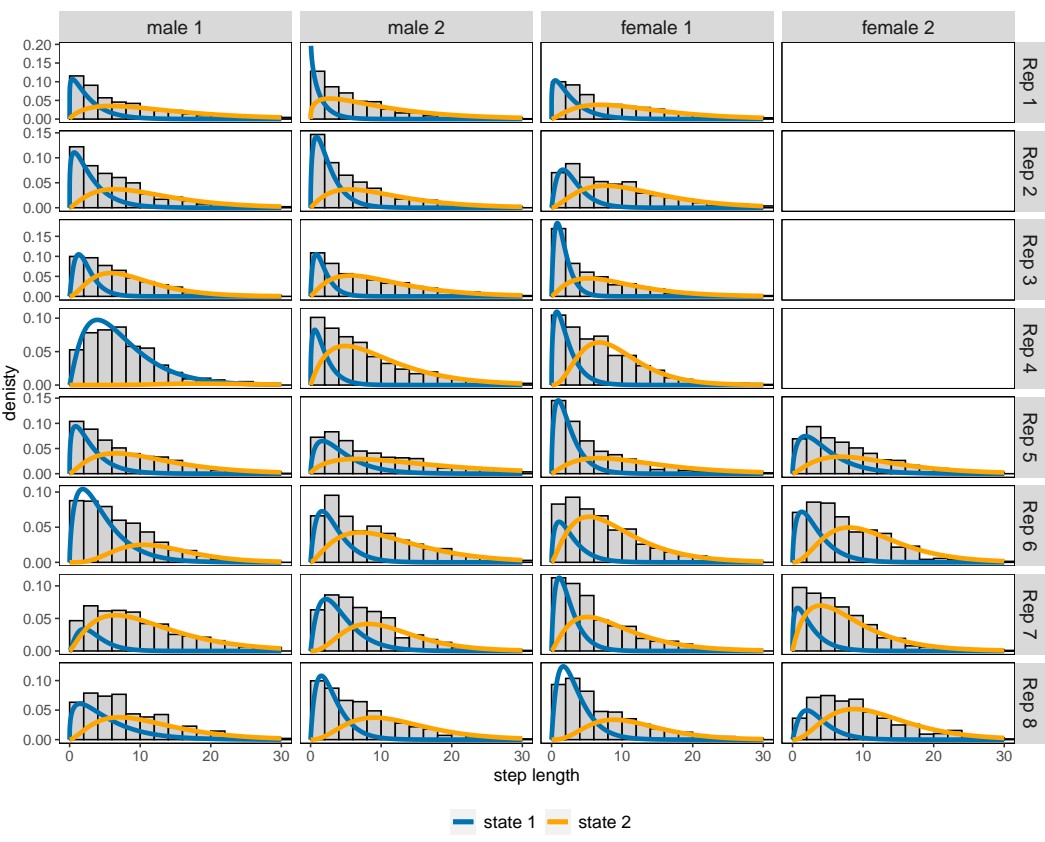

**Figure 4 Estimated state-dependent gamma distributions for step length as implied by the fitted 2-state HMM-iSSAs for each individual in replicates 1–8, respectively.** The distributions are weighted by the relative state occupancy frequencies derived from the Viterbi sequence. The gray histograms in the background show the distribution of the observed step lengths.

simulation scenario without state-switching, the HMM-iSSA was able to recovered the true underlying values in state 1, but produced unusable estimates in state 2 (Section S5.2).

## Case study on bank vole interactions

For most bank vole individuals, the HMM-iSSA approach could reasonably distinguish between two activity levels. State 1 was always associated to shorter step lengths compared to state 2 which could correspond to a rather inactive behavior (Fig. 4; mean of the estimated gamma distribution for step length ranging from 1.43 to 7.38 in state 1, and from 8.13 to 20.85 in state 2, respectively). According to the Viterbi decoded state sequences, the "less active" state 1 was occupied between 15.29% and 66.71% of the observed time period (Table S7), except for male 1 in replicate 4 which spent 96.43% of the time in state 1 according to its decoded state sequence. It is also the individual with the largest estimated mean step length in both states (7.38 in state 1, 20.85 in state 2). Thus, for this male, interpretation must be taken with care. The TS-iSSA provided mostly similar results for the movement kernel and state classification (Fig. S11).

For 21 of the 28 bank vole individuals, the HMM-iSSA results implied neutral behavior towards conspecifics in state 1 as all selection coefficients were non-significant ($\alpha = 0.05$; Fig. 5 and Fig. S10). This matches well with the interpretation of a less active/inactive state. For two individuals, the results indicated avoidance behavior in state 1. In state 2 ("active state"), most bank voles showed attraction to at least one bank vole of opposite sex as implied by the positive and significant selection coefficients. However, for four males and four females, the coefficients for occurrence of individuals with opposite sex were non-significant in both states. These are mainly the individuals for which the iSSA also implied neutral behavior (Fig. 5). However, for 3 individuals, *i.e.*, male 1 in replicate 7, female 1 in replicate 4, and female 1 in replicate 8, the HMM-iSSA indicated attraction towards another individual of opposite sex, while the iSSA indicated neutrality. The opposite is true for female 1 in replicate 7 for which only the iSSA indicated attraction. The selection coefficients for occurrence of individuals with same sex usually implied neutral behavior in state 1, and neutral or attraction behavior in state 2 (Fig. S10).

Overall, the results of the TS-iSSA are in line with the results of the HMM-iSSA (Figs. S10, S12), although the implications are slightly different for nine individuals. Regarding information-criteria based model selection, for most bank voles, AIC and BIC pointed to the hidden Markov step-selection model (Table S8). However, for 10 individuals, including half of the female individuals, BIC selected a simpler model, *i.e.*, iSSA or HMM. The selection of HMMs mainly corresponded to cases with many non-significant HMM-iSSA selection coefficients. The iSSA was preferred by BIC for male 1 in replicate 4 and female 2 in replicate 8.

# DISCUSSION

In this article, we discussed the relationship between standard iSSA without underlying behavioral states, the two-step approach TS-iSSA, which accounts for behavioral states *via* a classification of the movement data prior to performing iSSA, and the joint approach HMM-iSSA, which accounts for underlying behavioral states by combining HMMs and iSSAs in a single model. In particular, we compared the three approaches in both a simulation and a case study and highlighted possible consequences of either ignoring underlying behavioral states or using a prior HMM-based state classification to take them into account. This provides important implications for the practical application of fine-scale habitat selection analyses.

Combining ideas of iSSAs and HMMs in a single model, HMM-iSSAs build a convenient modeling framework to study state-dependent movement and habitat selection based on animal movement data (*Klappstein, Thomas & Michelot 2023*; *Nicosia et al., 2017*; *Prima et al., 2022*). This makes a prior state classification unnecessary, which, as demonstrated in the simulation study, could otherwise lead to biased estimates and misleading conclusions (see also *Prima et al., 2022*). In particular, the HMM-iSSA accounts for uncertainties in both the latent state and the observation process which allows for further inference, while the TS-iSSA completely ignores the uncertainties in the state decoding. This renders classical *p*-values of the TS-iSSA invalid.

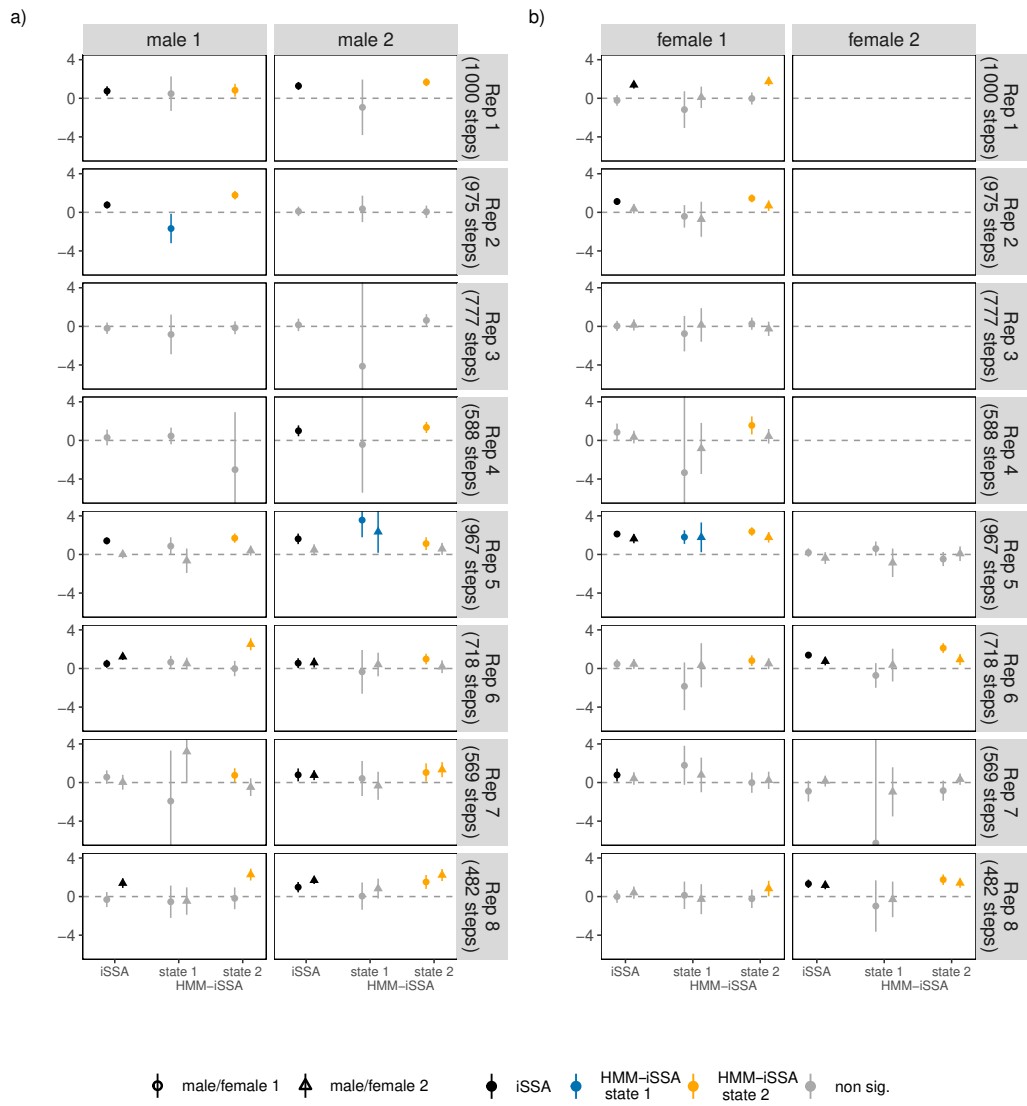

**Figure 5** (A–B) Estimated iSSA and HMM-iSSA selection coefficients (solid points/triangles) of inter-action behavior between individuals of opposing sexes within the eight replicates (1–8), including 95% confidence intervals (solid lines). Each replicate consisted of two males (male 1 and male 2) and one or two females (female 1 and female 2) such that each individual could respond to up to two opposite-sex individuals (dot: response to female/male 1, triangle: response to female/male 2 within a replicate). Non-significant coefficients (*p*-values below 0.05) are grayed out. The horizontal dashed line indicates zero (*i.e.*, neutral behavior); positive coefficients indicate attraction, while negative coefficients would indicate avoidance.

Moreover, the HMM-iSSA can detect states associated to same movement but different selection behavior (Scenario 2 in the simulation study), which is not possible using a prior classification that ignores the selection patterns. While Scenario 2 ("switching preferences") might cover a rather extreme case, one could imagine, for example, an underlying hungry and a thirsty state where the animal is searching for either food or water, or an attraction and neutrality/avoidance state where the animal is either attracted to another individual

or ignoring/avoiding possible social interactions. Even if the movement patterns might not be completely the same across these states, they might largely overlap and therefore lead to problems and high uncertainties in the prior state decoding of the TS-iSSA. This is contrasted with Scenario 3 ("HMM") of the simulation study which does not include any habitat selection, the states are solely associated with different movement kernels. Here, an HMM-based classification is suitable, and the TS-iSSA and HMM-iSSA perform equally well. Still, the TS-iSSA does not propagate the uncertainties of the state-decoding.

Although the main focus of the HMM-iSSA is habitat-selection analysis, it can also serve for behavioral classification. In this sense, it extends the classic HMM for state inference based on positional data (*Langrock et al., 2012*; *Patterson et al., 2008*) to account for habitat selection. However, various other data sources and classification methods have also proven useful for classifying animal behavior. Tri-axial accelerometer, which measure acceleration in three dimensions and provide information about energy expenditure, are particularly promising (*Nathan et al., 2012*). Here, especially supervised machine learning methods such as random forests and XGBoost show a good performance and allow for a fine distinction of behavioral categories (*Dentinger et al., 2022*; *Nathan et al., 2012*; *Sur et al., 2023*; *Yu & Klaassen, 2021*). However, supervised learning requires labeled data, *i.e.*, direct observations of animal behavior in the field or semi-natural enclosures, which are often not available. Unsupervised approaches, on the other hand, infer behavioral states completely data-driven without the need of direct observations. For accelerometer data, unsupervised K-means, mixture models or HMMs have, for example, been used in this context (*Leos-Barajas et al., 2017*; *Sur et al., 2023*). In general, such data-driven approaches require a careful interpretation of the inferred behavioral states, which also holds for the HMM-iSSA described in this article. If, on the other hand, information about the underlying states is available from auxiliary data or observations, the HMM-iSSAs can also be estimated in a (semi-)supervised manner.

It is straightforward to integrate other classification methods and additional data such as accelerometer data into the first step (prior state classification) of the TS-iSSA. This might improve its performance, although uncertainty quantification remains a problem. In principle, hidden Markov step selection models could also be extended to include additional data streams in the observation process, as briefly discussed in *Klappstein, Thomas & Michelot (2023)*. However, care must be taken that the resolution of all data streams matches and that the (in-)dependency assumptions of the model are appropriate.

Our simulation study demonstrates that ignoring underlying behavioral states completely by using standard iSSA can lead to biased and misleading results on selection behavior. While theoretically expected, a systematic evaluation and quantification of this effect had been lacking. Our study shows that iSSA tends to average out different selection behaviors in different behavioral states. This can lead to simple over- or underestimation of selection strength, keeping the overall direction of selection (*i.e.,* avoidance or attraction) correct. However, it can also lead to more serious problems when selection behavior has opposing directions in different states. In this case, we found that selection was estimated to be non-significant, which would lead to a strongly erroneous biological conclusion. This result corroborates the surmise that small effect sizes or non-significant results in

step-selection analyses may in fact be due to ignored underlying behavioral state switching (*Schlägel et al., 2019*) and more generally the caveat that failure to detect an effect does not necessarily imply lack of an affect.

While we have only considered fairly simple model formulations in this article, a major advantage of iSSA is the possibility to include covariate effects also on the movement kernel (*Avgar et al., 2016*). By using interaction terms between movement covariates (*e.g.*, step length) and environmental or temporal covariates, an iSSA can, for example, model the step-length distribution depending on time of day, snow depth or presence of roads (*Prokopenko, Boyce & Avgar, 2017*; *Signer et al., 2023*). Similarly, it is possible to include interaction terms of selection covariates (*e.g.*, landcover and distance to water) to allow for more flexible habitat selection patterns. In certain cases, it might thus be possible to capture some behavioral variability, *e.g.*, across environmental conditions or over time, in an iSSA with suitable interaction terms. But the resulting model will generally be more difficult to interpret and reflect the state-switching patterns described in this article (*i.e.*, abrupt pattern changes) only to a limited extent, unless the state process is observed or itself completely driven by an observed covariate such as "day or night".

Recently, *Klappstein, Thomas & Michelot (2023)* have extended the HMM-iSSA for the case that the latent state process, more precisely its transition probability matrix, is indeed influenced by other observed covariates. This can improve the biological realism of the model. Furthermore, *Nicosia et al. (2017)* discusses the inclusion of angular covariates to approximate a multi-state biased correlated random walk. Thus, HMM-iSSAs can be adapted to various situations and research questions. They have successfully been applied to study habitat selection of bison and zebra in encamped and exploratory states (*Klappstein, Thomas & Michelot 2023*; *Nicosia et al. 2017*; *Prima et al. 2022*), to detect the onset of mule deer migration and to evaluate the behavioral response of bison on the presence of wolves (*Prima et al., 2022*). In our case study, we extend the scope of application to fine-scale interactions of simultaneously tracked bank voles. Here, the 2-state HMM-iSSA provided a reasonable separation into a rather inactive state mostly associated with neutral behavior towards the conspecifics, and an active state often associated with attraction behavior. However, according to the decoded state sequences, the voles spend more time in the active state as expected based on *Mironov (1990)* (62.73% of the time on average). For one male bank vole individual, the state-classification within the HMM-iSSA was different. Its second state captured only rare observations with large displacement, while the first state accounted for all other observations. Here, the Viterbi sequence assigned over 96% of the observations to state 1 and the estimated HMM-iSSA showed larger mean step lengths in the estimated state-dependent gamma distributions than for all other individuals. Thus, the second state either captured rare events or outlying observations. This demonstrates that similar care is needed when interpreting the HMM-iSSA states as for general HMMs in an unsupervised learning context (*McClintock et al., 2020*).

In the active state, we generally expected males to look for females, while females might show different interactions with males depending on the reproductive state (*Schlägel et al., 2019*). For example, females in estrous may actively seek out males to generate mating opportunities away from the nest to lower the risk of infanticide (*Eccard et al., 2018*).

In contrast, females that are not in estrous state might show avoidance or neutrality toward males. In line with *Schlägel et al. (2019)*, for the male bank voles, we found either attraction or neutral behavior toward the females in the active state. This was, however, also the case for the female responses to male occurrences. While this might reflect the true individual interaction patterns, it might also be an artifact of measurement errors or the fence around the enclosures that limited the space use. Furthermore, some selection coefficient estimates had rather large confidence intervals possibly associated to the small number of observations for the rather complex model structure. This also prevented the use of a 3-state model that might have been able to differentiate between pure foraging and social interaction states.

With behavioral states being unobserved, it is usually unclear whether they manifest themselves in a given empirical data set. In both the bank-vole and simulation study, we therefore considered information criteria to select between the candidate models iSSA, HMM and HMM-iSSA. Especially the BIC performed well in our simulation study. For the TS-iSSA, such likelihood-based criteria cannot be applied as there is no proper joint maximum log-likelihood value for the state and observation process. This is another drawback of the two-step approach. Besides indicating if the inclusion of states or the inclusion of the selection function are appropriate for a given application, information criteria could also be used to select between HMM-iSSAs with different covariate sets or generally to select an appropriate number of meaningful biological states $N$. In the context of HMMs, however, the latter has proven difficult, as information criteria, especially the AIC, tend to select overly complex models with a rather large number of states (*Celeux & Durand, 2008*; *Pohle et al., 2017*). We expect this to be the case also for HMM-iSSAs. Therefore, besides information criteria, the selection of the number of states should further be based on a close inspection of the fitted models, and involve expert knowledge ("pragmatic order selection", *Pohle et al., 2017*). This is also highlighted in the supplementary simulation scenario which does not include any state-switching. Furthermore, future research could focus on the development of appropriate model checking methods for (Markov-switching) step-selection models.

It is important to note that the resolution of the data in time and space can strongly influence the model results and interpretation. Data sets with different resolutions might reflect different state, movement and selection patterns of an animal (*Mayor et al., 2009*; *Adam et al., 2019*). For example, an individual can exhibit many different behaviors during a long time interval, *e.g.*, during 24 h. Thus, a coarse time resolution might hinder the model to detect biological states such as resting and foraging or provide only crude state proxies. However, migration modes might be reflected in the data. On the other hand, movement and selection patterns might not directly be expressed in steps at very fine time resolution, *e.g.*, based on one location every second (*Munden et al., 2021*). Thus, the temporal resolution of the data must match the time scale in which the animal expresses its state, movement and selection patterns of interest. Moreover, if the spatial resolution of a covariate map is too coarse, important habitat features might be overlooked in the analysis (*Zeller et al., 2017*). Thus, the resolution of the data is a key factor in HMM-iSSAs. However, once movement and habitat data are available at a suitable resolution in space

and time for a given species and research question at hand, the HMM-iSSA approach can flexibly be applied to study fine-scale state-dependent movement and habitat selection.

In our study we systematically compared the performances of two methods to incorporate behavioral modes into step-selection analysis with the approach to simply ignore it. We found that the more complex yet more elegant way to use the combined HMM-iSSA yielded considerably better results. Therefore, to facilitate its use, the basic HMM-iSSA is implemented in the R-package HMMiSSA available on GitHub (*Pohle & Signer, 2023*). With this implementation, we hope to encourage the method's wider application in habitat selection studies. Obtaining more accurate estimates of habitat selection will improve our understanding of the driving forces of animal movement as well as predictions of space use, and may thus ultimately serve conservation efforts such as the planning of protected areas or movement corridors.

## ACKNOWLEDGEMENTS

We thank Sophie Eden, Angela Puschmann and Pauline Lange for help with the bank vole data collection and maintenance of the outdoor enclosures. We thank the three anonymous reviewers for their comments and suggestions which helped to improve the manuscript.

### Funding

This work was supported by the Deutsche Forschungsgemeinschaft (DFG, German Research Foundation, grant no. SCHL 2259/1-1). The APC was funded by the Deutsche Forschungsgemeinschaft (DFG, German Research Foundation) - Projektnummer 491466077. The funders had no role in study design, data collection and analysis, decision to publish, or preparation of the manuscript.

### Grant Disclosures

The following grant information was disclosed by the authors:
Deutsche Forschungsgemeinschaft (DFG, German Research Foundation: SCHL 2259/1-1.
Deutsche Forschungsgemeinschaft (DFG, German Research Foundation): 491466077.

### Competing Interests

The authors declare that there are no competing interests.

### Author Contributions

- Jennifer Pohle conceived and designed the experiments, performed the experiments, analyzed the data, prepared figures and/or tables, authored or reviewed drafts of the article, and approved the final draft.
- Johannes Signer conceived and designed the experiments, authored or reviewed drafts of the article, finalised the R-package, and approved the final draft.
- Jana A. Eccard performed the experiments, authored or reviewed drafts of the article, and approved the final draft.

- Melanie Dammhahn performed the experiments, authored or reviewed drafts of the article, and approved the final draft.
- Ulrike E. Schlägel conceived and designed the experiments, authored or reviewed drafts of the article, and approved the final draft.

### Animal Ethics

The following information was supplied relating to ethical approvals (i.e., approving body and any reference numbers):

The Landesamt für Umwelt, Gesundheit und Verbraucherschutz Brandenburg and the Landesumweltamt Brandenburg provided full approval for this research.

### Data Availability

The case control dataset used for the case study and the landscape rasters used for the simulation study: Pohle, J., Signer, J., Eccard, J. A., Dammhahn, M., & Schlägel, U. E. (2023). Data and R-Code from: How to account for behavioral states in step-selection analysis: a model comparison [Data set]. Zenodo. Available at https://doi.org/10.5281/zenodo.10101651

The raw data is available at Dryad: Schlägel, Ulrike E. et al. (2019). Data from: Estimating interactions between individuals from concurrent animal movements [Dataset]. Dryad. Available at https://doi.org/10.5061/dryad.rt535m8.

The R Code is available on the Zenodo repository (https://doi.org/10.5281/zenodo. 10101651).

### Supplemental Information

Supplemental information for this article can be found online at http://dx.doi.org/10.7717/ peerj.16509#supplemental-information.

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
