# Peer review of "How to account for behavioral states in step-selection analysis: a model comparison"

_PeerJ, doi:10.7717/peerj.16509_

## Round 0.1 · original submission · Minor Revisions

Three expert reviewers found much to like about your paper, but all provided suggestions for improvement. I agree with Reviewer 3 that care should be taken to diligently connect your current work to all relevant previous work. Note that a reviewer suggests citing a new paper that may not have been published at the time your manuscript was prepared. It looks to me like all the suggestions have to do with writing and interpretation, so a decision of minor revisions seems appropriate. I encourage you to consider all suggestions, especially those from Reviewer 3.

Reviewer 1 ·

Basic reporting

Reporting is clear and the background provided is good. Results are relevant. The provided code can be opened and the links to the data in drobox work. I have not ran all of the code.

Experimental design

Original and clear design and meaningful. The authors clearly explain why each part of the research is conducted throughout, apart from a few small places which I have provided in comments. The investigation is rigorous, particularly regarding the simulation and comparison of methods. I agreed to review this paper as I intend to use the methods at some point and it is clear how to replicate this.

Validity of the findings

No comment - this is generally good.

Additional comments

Dear Authors,

Thank you for your compelling and comprehensive work. I really enjoyed both the concept of the modelling framework and the presentation of the simulation study when compared to the two-step iSSAs and also the ‘one-step’ iSSA. I believe this manuscript will be of high impact and am I very likely to adopt the MS-iSSA techniques into my own step-selection analyses in the future. I believe that the comparison with iSSA is particularly powerful and I believe that the way it is communicated, alongside the provided R package will be successful in convincing researchers to adopt procedures such as this one which incorporate different behaviors into the movement analyses.

As I am familiar with Sclhagal (2019), it was good to see the bank vole data set used to test out a real world comparison of iSSA and MS-iSSA. It would have been good to see this applied to animals that are not moving in response to complex social behaviors to be able to present it in a more simple manner. However, I believe that the simulation study is very clear and has a pedagogical advantage over most research articles.

I found little fault with the entire manuscript after reading it through several times and therefore I provide only minor comments below which are generally focused on the communication of methods. As an overall comment you should consider checking through the manuscript to ensure you have explained what every single variable or parameter means, as I may have missed instances where they are not explained. Furthermore, I have checked the author checklist which I believe to be sound. However, I have only read the supplementary materials once, therefore you may wish to check through the SI regarding clearly explained mathematical notation. I provide some minor comments below.

Introduction:

Line 85 – What exactly do you mean by “state decoding”

Methods:

Overall – Your equations are not numbered.

Line 104 - (the mathematics) what is the 0 subscript? Later you mention it to be m=0,…,M but what does this mean here?

Line 109 – add an s to location

Lines 103 – 117 are very well explained otherwise.

Line 121 – (the equation below) Should \tilde{x} really be within the set of all real numbers? Should you not be integrating over the entire available landscape? This also occurs in further equations. I believe you should use another letter to define the landscape.

Line 121 and elsewhere – (the equation below) You should define what all the variables are with a tilde. I suspect they are functions of \tilde{x} rather than x, but this should not be assumed.

Line 122 - /Phi should be /phi I think

Line 128 and elsewhere– (the equation below) You are taking the dot product of the two vectors within the exponential function therefore you should use a \cdot

Line 151 – (the reference to the reduced Equation) This is also well explained in Avgar (2016).

Line 168 – You mention a “suitable” distribution for the turning angles and step lengths, what does this mean? Would you not use the distributions you defined above for the movement kernel to sample the movement steps?

Line 187 – I think you missed some commas from the p_0ti

Line 190 – is this m the same subscript as the 0 in the first line of the methods section?

Line 243 – It doesn’t really seem necessary to have used three different amounts of control steps and it may make your Figure 3 easier to read if you just focus on M=500, but that is only my opinion.

Lines 252 & 302 – What R packages were used for the simulation study?

Line 267 – What does 602-1 mean?

Results:

Figure 3 – This is a nice powerful figure. Could you denote the left y-axis with the appropriate letters rather than the words?

Lines 304 – 307 I think starting off with talking about the different amount of control steps having little effect isn’t a very interesting way to start off the results section. You should consider instead starting with a sentence that explains how well the MS-iSSA model does in the simulation study, in comparison to the others, and then talk about the details.

Table 3 – Can you use % somewhere to indicate what the values are

Line 343 – Do we have any information on the breeding ‘state’ of the Bank Voles in the model? As you previously mentioned, the attraction between the sexes generally is due to that.

Figure 4 – Are the gray histograms for state 1 or overall steps? Could you split these into the state 1 and state 2 steps and have overlapping histograms?

Discussion:

Line 374-375 – If a reader skips to the discussion they might not understand the TS-iSSAs and MS-iSSAs, could you remind the reader what these are?

General – I think the discussion is great, and I found each point you made to be suitable and interesting. However, I would like to see more discussion on the available methods of animal movement analysis and behavior classification methods outside of the SSA and HMM frameworks. Perhaps also on behavior classification validation.

Lines 465-466 – Its fantastic that there is an R package available. I think you really should summarize your findings here whilst announcing this. It seems a it of a disappointing end to a good discussion.

Reviewer 2 ·

Basic reporting

I found the paper to be extremely well written, informative, and well organized.

The authors do a nice job reviewing the literature, but they missed an important and highly relevant reference by Klappstein et al. (2023) that was likely drafted independently right around the same time that the authors were working on this paper:

Klappstein, N. J., Thomas, L., & Michelot, T. (2023). Flexible hidden Markov models for behaviour-dependent habitat selection. Movement Ecology, 11(1), 30.

Although there is some overlap between the authors’ paper and Klappstein et al. – both highlight the potential benefits of the MS-iSSA approach – this paper (by Pohle et al.) makes several important and unique contributions: 1) the authors demonstrate differences between popular analytical methods (i.e., ISSA, HMM, two-step ISSA, state-switching ISSA) using an in-depth simulation study; and 2) they revisit an important analysis by Schlagel et al and test the hypothesis outlined in that paper that the lack of statistically significant selection coefficients might be explained by behaviorally mediated habitat selection, for which they found some support.

It would be appropriate to cite Klappstein et al. in a few places (e.g., lines 77-81; line 381, and possibly line 410).

The authors provide all data and code used to reproduce the analyses in the paper along with associated documentation. This was much appreciated!

Experimental design

The authors’ simulation study was well designed for highlighting the differences between the analytical approaches considered in the manuscript (i.e., ISSA, HMM, two-step ISSA, or state-switching SSA) and the potential advantages of the MS-iSSA approach when individuals exhibit behaviorally mediated habitat selection, which include:

1. Less biased estimates and better control of type I error rate when using the MS-iSSA approach (compared to the two-step iSSA).
2. Better state classifications than the HMM that ignores habitat selection.
3. Unbiased estimates of selection parameters when movement kernels are the same in all behavioral states but selection parameters vary by state; in this situation, the MS-iSSA is unreliable.
4. The ability to use information theoretic approaches for model selection, which is not possible when using the two-step iSSA.

The authors’ case study also provides a nice comparison of the methods using real data and allowed them to test the hypothesis that the lack of statistically significant selection coefficients in a previous iSSA might be explained by behaviorally mediated habitat selection, for which they found some support.

The authors describe their methods in detail, and provide additional information in supplementary appendices.

Validity of the findings

The authors’ simulation study and applied analyses appear sound and their results are in line with what I would expect to see. They used sufficient replicates in their simulation study to be able to draw meaningful conclusions and explored sensitivity of results to potential Monte Carlo error (by repeating analyses with different numbers of available points). The conclusions are well stated and sound.

Additional comments

I have a few minor (mainly editorial) suggestions, below:

1. Line 50: suggest distributions (plural)
2. Line 52: I think it is a bit odd to say that movement is “restricted” by habitat selection – consider influenced?
3. Line 109: suggest locations (plural)
4. Line 129: suggest deleting “value” to read, “Where a positive selection coefficient indicates preference for, and a negative *coefficient* avoidance of…”
5. Line 143: suggest “assuming step lengths follow”
6. Line 145: suggest “turn angles follow”
7. Lines 143-160: this was extremely well written!
8. Line 186 (and more generally): I would stick to Markov-switching iSSA rather than refer to a state-switching conditional logistic regression (CLR). CLR is just a trick to maximize the (MS-) iSSA log likelihood.
9. Line 256: it might help to define what a “replicate” is when this term is first introduced.
10. Line 263: I would drop the decimals associated with average home range size and core area (this seems like excessive precision).
11. Lines 269-270: I think this sentence could probably be deleted “For the release day we choose dry and mild weather.”
12. Line 279: I would suggest using past tense “were computed”
13. Lines 299-302: it was not clear to me why the 2-state TS-iSSA required use of a “similar [but not completely equivalent] case-control data set for each individual.” Can the authors provide a short explanation for why they did not use the same data set when using that approach?
14. Line 302: I am not sure I understand what is meant by “we did not use any blinding methods.”
15. Line 399: rather than “corrupt” – perhaps rephrase in terms of providing misleading conclusions?

Reviewer 3 ·

Basic reporting

Fair, please see General Comments for details (some problems with presentation, references, and small issues with tables).

Experimental design

Very good, please see general comments for details.

Validity of the findings

Very good, please see general comments.

Additional comments

This paper explores the performance of a recent model; the HMM-SSF (or the MS-iSSA as the authors refer to it), a state-switching step selection model that characterises behavioural states based on movement and habitat selection. This is a very exciting model in movement ecology with many potential applications, and therefore, it is important to fully understand its estimation and predictive performance (i.e., how accurately it can estimate parameters and predict underlying states), and its potential improvements over two-stage approaches (e.g., better uncertainty quantification). I find the manuscript interesting, well-written, and full of useful information (in both the main-text and the appendices). I don’t have any serious concerns with the analysis, as I think it is generally well-done. As I see it, the main issues with the paper are related to the presentation of the model, and my remaining comments are relatively minor. Overall, I found it to be a good paper and I recommend its publication.

MAJOR COMMENTS:

1. This is not a technical issue of the paper, but I think it’s rather important: I strongly recommend using the established nomenclature for the model. It was coined the HMM-SSF by the original authors (Nicosia et al. 2017) and has been referred to as such in all follow-up papers. My understanding is that the model in your paper is an HMM-SSF as defined by Nicosia et al. (i.e., it is a state-switching SSF with movement and habitat variables), and you use the same suite of fitting techniques (i.e., likelihood approximation via random sampling, MLE, parameter corrections for non-uniform sampling). I don’t see a good reason to rename the model (and the associated analysis techniques), as this suggests that your model is different from that of Nicosia et al., and I am concerned that this will confuse many readers. This is particularly true in a field already inundated with potentially confusing terminology and nomenclature. I strongly suggest using the original terminology (HMM-SSF or HMM-SSA if you prefer to reference the analysis, rather than the model itself) to maintain consistency throughout the literature. In the absence of such a change, you should clearly demonstrate the key differences that warrant such a change.

2. In general (and not specifically to your paper), I have often found the distinction between SSA and iSSA to be somewhat ambiguous. I realise that you did not coin the term iSSA, but you are placing an emphasis on this distinction throughout your manuscript, and as such, I wonder if you could elaborate. For example, in the first few lines of your paper, you mention SSA and iSSA as distinct terms. Could you please clarify the differences? You state several times that iSSA is based on conditional logistic regression, but would maximising the full likelihood presented in Forester et al. 2009 (Equation 8) be considered iSSA (as it is no longer CLR)? Is it only the specific implementation methods (i.e., CLR implementation with parameter corrections) described in Avgar et al. 2016 (and also Forester et al. 2009)? I recognise that it is not your job in this paper to clarify these questions in the broad context of SSA literature, but I implore you to consider your use of terminology and be explicit about your definition of SSA/iSSA if you feel it is important to use such distinctions.

3. I agree that a state-switching model is extremely useful to capture heterogeneous patterns of movement and habitat selection, and I agree with many of the points made in lines 398-408. However, I wonder if the current phrasing may underplay the flexibility of SSFs, which can have very complex formulations not explored in this paper. For example, in scenario 1, you could allow habitat selection to vary with movement or time to capture some of the variability due to state-switching. I imagine that a state-switching model has interpretability benefits over an SSF with complicated interactions, but I think the flexibility of SSFs could be briefly discussed.

4. I think it should be clear from the beginning of your methods that this the same model as in Nicosia et al. 2017 (i.e., you should include a citation much earlier in this section). Without this reference, it is somewhat unclear if the model in your paper is different from previous HMM-SSF papers, which may be confusing to readers. Also, another recent paper (Flexible hidden Markov models for behaviour-dependent habitat selection. 2023. Movement Ecology, 11:30) is relevant to your manuscript and should likely be cited, particularly since there are so few former HMM-SSF papers.

5. Are iSSA-style parameter “corrections” (i.e., updating the tentative movement parameters; Avgar et al 2016) actually needed in your implementation? From my understanding, these corrections are needed when the analysis is being shoe-horned into a conditional logistic regression framework (and therefore, fit with CLR software that does not account for the sampling procedure). Nicosia et al. 2017 and Prima et al. 2022 used clogit() within their EM algorithm, and so these corrections were necessary. However, since you are implementing the likelihood with custom code, shouldn't it be possible to account for the sampling procedure directly, rather than post-hoc (i.e., a similar likelihood as in Forester et al. 2009)? The benefit being that parameter corrections are unnecessary, the downside being that it deviates from iSSA papers. Could you explain this implementation choice?

MINOR COMMENTS:

Line 63: Could you please cite the relevant software.

Equation 1: Can you please check the domain of integration (I believe it should be R^2 not R).

Line 79-81: Arguably, this is not “similar as Nicosia and Prima” but “the exact same as Nicosia and Prima”. I suggest re-wording to convey that this is the same model and changing MS-iSSA to HMM-SSA to match previous papers.

Lines 121-122: I realise that the terms “selection-free movement kernel” and “movement-free selection function” have been used in previous papers. However, I think this terminology is misleading, as the processes are not actually independent. I understand that you chose these terms because, if the habitat were perfectly homogeneous then the animal’s movement process would be entirely determined by phi. Conversely, the space use of an animal with no movement constraints would be entirely driven by omega. However, in practice, this phrasing gives the misleading impression that the model formulation used for one doesn’t affect the other, which will usually not be the case. For example, the choice of step length distribution might well have an effect on the estimated habitat selection parameters. I suggest rephrasing.

Lines 134-142: This is an interesting (and well-described) point!

Line 152: Worth noting that the coefficient for cos(angle) can paramaterise von Mises distribution with a mean of zero or pi (positive or negative beta, respectively), which is useful in state-switching models (see Nicosia et al. 2017).

Equations 3 and 4: Could please clarify why the step density includes the -log(l), but the state-dependent choice probability does not?

Lines 198-200: It’s a good idea to constraint the parameter estimation, and something that is missing from SSA in general.

Line 295: Did you find that you needed 500 locations to get a good estimate? This is different than the results of your simulation, and I imagine could be due to the complexity of your model, amount of data, etc. Could you clarify/discuss this choice?

Line 314: Should this read 39-42% (to match table 2)?

Table 2: I presume the iSSA find significant results in scenario 1 because it is detecting the selection in state 2. At first glance, this table seems to indicate that iSSA always detects an effect when there is none (which isn’t really the case). For the iSSA, I would consider centering this column under both beta values to indicate that the iSSA is modelling both simultaneously, rather than suggesting that it is only based on beta_1. The same is true for the TS-iSSA (scenario 2).

Lines 415-416: Why did you expect 25%?

Lines 434-435: Did you consider pooling all males/females to try a 3-state model?

Table S1: The “parameter” column suggests that each row corresponds to that parameter, which is not always the case. I suggest removing this column and replacing the distribution column with notation that contains the distribution parameters, e.g., Exp(lambda_i), Gamma(k_i, r_i). This will make it clearer that you are showing SSF parameter based on its relationship to the movement distribution and proposal distribution parameters. You are also missing I subscripts on the lognormal.

---

## Round 0.2 · accepted · Accept

Thank you for considering all reviewer comments and implementing them as appropriate. I assessed the revision myself without sending it back out for review because the changes were very well documented and referenced to the reviewers' minor comments. I enjoyed reading this revised version and I am excited to see it published. Very nice work!